# Deep Groundwater Flow Patterns Induced by Mine Water Injection Activity

**DOI:** 10.3390/ijerph192315438

**Published:** 2022-11-22

**Authors:** Ge Chen, Zhimin Xu, Dmytro Rudakov, Yajun Sun, Xin Li

**Affiliations:** 1School of Resources and Geosciences, China University of Mining and Technology, Xuzhou 221116, China; 2Department of Hydrogeology and Engineering Geology, Dnipro University of Technology, av. Dmytra Yavornytskoho, 19, 49005 Dnipro, Ukraine

**Keywords:** Liujiagou Formation, mine water injection, deep groundwater flow, storage capacity

## Abstract

Mine water injection into deep formations is one of the effective approaches for reducing the drainage from coal mines in the arid and semi-arid region of the Ordos basin, China. Many coal mines are attempting to execute the related projects. Under the influence of groundwater protection, the understanding of regional groundwater flow is becoming highly important to the mine water monitoring, whereas quite few academic research teams focus on the deep groundwater flow pattern by mine water injection. This paper reveals the spatial distribution of Liujiagou Formation that is in positive correlation with the terrain, and its local thickness is influenced by the dominant W-E and NE-SW directions of geological structures. Only a part of sandstone rocks consists of aquifers, the rest 61.9% of relatively dry rock provide the enhanced storage space and partial mudstone aquicludes decrease the possibility of the vertical leakage for mine water. The dynamic storage capacity is evaluated at 2.36 Mm^3^ per 1 km^2^ and over 25.10 billion m^3^ in this study area. Two hydrogeologic cross-sections of basin-scale identify the W-E and N-S regional groundwater flow directions, with the lower Yellow River catchment becoming the discharged region. The hierarchically and steadily nested flow systems containing coal mining claims are influenced by coal mining activity. The groundwater depression cone in a shallow coal measure aquifer is caused by mine water drainage whereas the groundwater mound in Liujiagou Formation is generated by mine water injection activity. The numerical simulation revealed that the groundwater head rebound is slightly decreased and will not recover to its initial baseline within 500 years due to its low porosity and permeability. This study elucidates the deep groundwater flow patterns induced by mine water injection and provides a practical methodology for the management and pollution monitoring of mine water injection activity.

## 1. Introduction

Groundwater is an important part of the global hydrologic cycle [1]; it is growing in demand in arid and semi-arid regions, especially in China. Mine water drainage [2] from coal mining areas, such as the Ordos Basin, Tarim Basin and Junggar Basin, aggravates this situation. Meanwhile, mine water is considered as one of the non-traditional groundwater resources, which is especially important during extreme events. For this reason, mine water deep transfer and storage [3] as one of methods for groundwater resource management has started to be applied in some coal mines of Ordos Basin. The general characteristic of alkaline mine water [4] often shows the hydrochemical SO_4_-Na type, with high TDS content up to 2400 mg/L and pH value at 7.84. It is common in the eastern margin of Ordos Basin. The deep storage aquifer of Liujiagou Formation shows a very high TDS concentration at 65,111 mg/L with the hydrochemical Cl-Ca·Na type and pH value at 5.35. Therefore, the approximative “Brackish Water Injection and Storage” of treated mine water is practicable for brine aquifers and it is interesting to quantify the geochemical evolution of groundwater induced by mine water injection. Additionally, a large number of oil-gas field injection wells are distributed in this region, and their successful application is also useful for mine water injection. Several million cubic meters of mine water has already been injected into the deep and tight sandstone voids. Mine water injection has been feasible by the current technique. Until now, however, deep groundwater flow patterns and geochemical evolution induced by mine water injection activities has not ever been properly studied. Deep groundwater flow regime and complex water-rock interaction are the fundamental challenge for hydrogeologists.

Due to the limitations to approaching and investigating deep groundwater systems, the results of studies of deep groundwater (>1500 m) flow evolution are poorly reported. Mine water is a niche area of research. Previous studies did not refer to the field of mine water and focused mainly on regional contaminant identification [5], oil accumulation [6], climate warming [7], groundwater age [8] and heat [9]. Tóth was the first to study basin-scale groundwater circulation, showing the analytical understanding by the cross-sectional flow field [10]. Two-dimensional water table-induced flow models were extensively applied to delineate the regional groundwater regime and circulation [11]. The regional groundwater flow is critical for the description of convective heat transport in basins; they are frequently adopted together to estimate the pathways and migration for each other, such as density and age change by the basin-scale groundwater temperature distribution [12]. Free thermal convection is critical for studying of the relationships between topography-driven groundwater movement and heat transfer. Water table variation is usually regarded as the dominant driver of regional groundwater movement. Moreover, a synclinal structure was found favorable for stimulating deep groundwater circulation on its two wings [13]. Springs are used to diagnose the hydrogeologic processing in basin-scale region [14].

Furthermore, the research focusing on water injection for the unconventional oil and gas recovery [15] at Williston Basin of Canada demonstrated that a large amount of produced and injected fluids have changed the fluid flow rates and directions at the sedimentary basin scale. A study of large-scale geologic storage of CO_2_ in deep rock formations at Tokyo Bay showed that carbon dioxide storage activity has caused considerable pressure perturbation and significantly affected the regional groundwater system. Groundwater pressure buildup, migration to the land surface and discharged to the shallower aquifers is examined [16]. The degree of permeability heterogeneity controls the groundwater flow system distribution [17].

Deep groundwater characteristics not only indicate the valuable information of hydrosphere, but also an effective tracer for groundwater evolution including dissolution, sedimentation, evaporation and mixing [18]. According to the statistical information of deep groundwater quality from the southwestern of United States, the results reveals that the usable groundwater (TDS < 10,000 mg/L) holds 38% in the Great Basin and some TDS concentrations are up to 523,977 mg/L [19]. The slow-flow or stagnation of deep aquifer will reflect high TDS [20]. Sometimes, the rare-earth pattern and isotopes would indicate a large-scale lateral groundwater flow or inter-aquifer flow in the depths exceeding 2000 m [21].

However, how mine water injection activities influence the deep groundwater flow and geochemical evolution have not been studied by the previous literature. Understanding this basic theoretical research in the field of mine water injection will be the fundamental advancement and significant progress.

In this study, collected drill hole information, geophysical technique and numerical simulation are proposed to identify the spatial distribution, horizontal and vertical heterogeneous characteristic of Liujiagou Formation. The original deep regional groundwater flow cross sections that are disturbed by mine water drainage and injection activities will be elaborated to describe the effect on the regional groundwater flow pattern and the long-term rebound of the groundwater head to its initial baseline. The industrialization management for mine water injection in the coal mining claims would be a novel expectation.

## 2. Description of the Study Area

The study area is located in the eastern margin of Ordos Basin, which is the ecologically fragile area in the western arid and semi-arid region of China. It is rich in coal, trona, mirabilite, oil and gas resources. Many large coal mines with high production are located here. A total of 1.223 billion tons of coal was produced in Ordos city and Yulin city in 2021. As shown in Figure 1, most parts of the study area are layered on Yimeng Uplift and Yishan Slope, the stratum is approximatively horizontal. As stated in the introduction, the high frequency coal mining activities in Jurassic Formation have induced a large amount of groundwater to be overexploited and drained to the surface. Due to the little rainfall and arid environment, increased groundwater depletion resulting from mining activity leads to water scarcity and adverse ecological impacts on the coal mining claims [22]. According to the statistical data, 1.2 billion tons coal was mined out in 2021 and 2.5 billion cubic meters of groundwater was pumped. Compelled by the severe situation, groundwater storage recovery in deep aquifers by mine water injection activity has started to be operated in Muduchaideng, Nalinhe 2 and Tangjiahui coal mines to restrain the regional groundwater loss rate [3]. In this paper, the large scale of the eastern margin is selected as the targeted region to study the deep groundwater flow patterns in Liujiagou Formation (LFm) [23] of the Lower of Triassic. LFm is not well-covered by publicly available papers due to no occurrence of natural resource.

LFm is the lower of Triassic and characterized with little porosity, low permeability [24] and high content of quartz (average 40.1%) and feldspar (average 31.1%). The lithology of LFm is interlayered with different thickness of purple mudstone, grey-white and flesh red sandstone [25]. Mine water from Jurassic coal measures [26] and its overlying aquifers is injected into LFm by high pressure pumps at surface, as shown in Figure 2.

## 3. Mathematical Model and Simulation

Two dimensional simulations of different cross-sectional models through the basin-scale could be governed by both hydraulic water-table and stream function in the steady flow equations [27] is expressed as follows:(1)$$\frac{\partial }{\partial x}(K_{x}\frac{\partial h}{\partial x})+\frac{\partial }{\partial z}(K_{x}\frac{\partial h}{\partial z})=0$$
(2)$$\frac{\partial }{\partial x}(\frac{1}{K_{z}}\frac{\partial \psi }{\partial x})+\frac{\partial }{\partial z}(\frac{1}{K_{z}}\frac{\partial \psi }{\partial z})=0$$
where $K_{x}$ and $K_{z}$ are the hydraulic conductivity (m/d) along Ox and Oz axes, respectively; h denotes the hydraulic water table or head (m); ψ represents the stream function and its isolines shows the streamlines of a flow net. The hydraulic water head h can be extended with the Girinski potential for groundwater flow in stratified aquifers [28], which allows for the application of Equations (1) and (2) to the rocks with layered heterogeneity.

The surface of the finite element mesh is approximately consisted with the groundwater table generally following the topography [29] with the head boundary condition. In this model, the water depression cone resulting from climate change and epeirogenetic geology were not taken into consideration. The detailed configuration of the hydrogeological condition is described in Section 4.5.

## 4. Result and Discussion

### 4.1. Spatial Information of Liujiagou Formation

According to the data mining and analysis for the identified 37 wells from salt and coal mines, oil and gas fields, we generalized the spatial geodata in the study area of LFm as shown in Figure 3a,b. The overall trend indicates that the inclination from the east to the west due to its location on Yishan Slope [30]. The top boundary elevation of LFm ranges from −1287 m to 615 m, the bottom boundary elevation ranges from −1642 m to 402 m with its thickness of 115 m to 465 m. By the domination of whole and small-scale fluctuation of layers, the LFm appearance was inclined from east to west. The eastern and western depth was up to 700 m and 2700 m, respectively. Under the predominantly horizontal strata tendency, some distinction rules were identified, e.g., the northern zone located on the Yimeng Uplift [31] had a relatively small thickness of 115–250 m, whereas the LFm thickness in the southeastern and western zone located on the Yishan Slope is stable and continuous.

### 4.2. Regional Heterogeneity Characteristics of the Formation Distribution

Geophysical exploration methods including 3D seismic, gravity [32], magnetic and electrical tools [33] were generally used to investigate the characteristics of structure, bedrock interface and so on. The regional Bouguer gravity and magnetic anomaly weremapped in [34] for the stable cratonic Ordos basin. Reduction-to-pole for magnetic anomaly, medium and small scales of residual gravity anomaly were presented to describe the generally morphological characteristic of LFm, as shown in Figure 4. These geophysical explorations were treated as one useful method to distinguish the horizontal heterogeneous characteristic in the ungauged region due to the stable strata during the Paleozoic to the Mesozoic of Ordos basin.

The comparison of Figure 3a and Figure 4a shows that the Bouguer gravity anomaly is positively related to the top interface of the LFm elevation. The lithology of LFm consists of low-permeable sandstone interlayered mudstone, and its density was higher than the covering formations from the measured compensating density data of well logging in the MC-1 well as shown in Figure 5. The large Bouguer gravity anomaly in yellow color indicates the shallower depth of LFm in comparison to the western region in green color, which is located in the deeper zone of Yishan Slope. Through the combination analysis of Bouguer gravity anomaly and reduction-to-the-pole of magnetic anomaly, the large-scale inferred faults in purple dash lines were obtained (Figure 4). On the whole, the quantity of faults is relatively smaller; they are mainly concentrated in the zones of gradient or extreme values of gravity and magnetic anomaly [35]. When we focused on the local small-scale residual gravity anomaly in Figure 4c, the horizontal heterogeneous characteristic was more obvious. In contrast to Figure 3b, large-scale faults may disconnect the deeper Proterozoic strata [36] and local small faults control the LFm thickness variation. The banded distribution is seen in two dominant directions, the first one is W-E and the second one NE-SW [37], which is consistent with the base faces of fluvial meandering channels [38], but differs from the S-N and WN-ES directions of faults strike that were ascertained near the injection well by 3D seismic exploration from the CCUS project field.

Therefore, the combination of drillhole, gravity and magnetic methods were able to describe the regional thickness and banded heterogeneous characteristics for LFm. The dominant W-E and NE-SW directions may influence the groundwater flow direction and the circulation rate. The faults zone with the derived fractures has a relatively high porosity and conductivity for mine water injection.

### 4.3. Vertical Heterogeneity Characteristics of LFm: Taking the MC-1 Well as the Example

Based on the few available papers, drill holes and the high cost of deep exploration, as well as the approximately horizontal geologic structure in the study area, we took the MC-1 well as the example to characterize the vertical heterogeneity of LFm.

According to the geophysical log as shown in Figure 5, the red density curve of LFm reveals the tight strata depths. As the buried depth increased, the density affected by the geologic compaction was aggrandized [39], which has the same trend as the geothermal gradient. The average density of LFm was 2.42 g/cm^3^, the high density means that its sonic properties are lower than in other formations due to low porosity and fracture ratio. When focusing on the green natural gamma curve, a relatively higher content of argillaceous mud may enhance this performance due to the main source of gamma-ray from the minerals of mud fraction [40]. The average natural gamma of LFm was up to 110 API; therefore, the well diameter in the section of LFm implied that some segments of mudstone contributed to the corrosion. The high argillaceous mud of LFm was also responsible for the high self-potential, with the value being higher than other formations overall. The fluctuant curve of shallow/deep lateral log is consistent with that of Yan’an Formation, but their difference indicates that the fragility characteristic was abundant with vertical fractures and burrows [41], which was verified by cores and lab tests.

By combining the systematical analysis of the above geophysical log, 35 original sub-aquifers were identified. We evaluated the thickness of sandy mudstone, medium sandstone, interlayered siltstone, fine sandstone, coarse sandstone, mudstone and siltstone that are 27.53%, 23.97%, 21.28%, 13.81%, 6.52%, 4.24% and 2.65% of the whole LFm thickness, respectively. Not all aquifers have the lithology of sandstone with particle granularity over fine stuff; only 38.1% of LFm was identified as the watered rocks. Dry sandstone containing gravitational water was not detected [42]. The statistic treatment showed that all of the coarse sandstone, 80.8% of medium sandstone, 38.7% of fine sandstone, 29.3% of siltstone, 14% of interlayered siltstone can be identified as the aquifers. From this favorable view, more space could appear in these dry rocks to store mine water when during high-pressure artificial injection activity [43].

Just for the vertical aquifers shown in Figure 5, different thick aquifers could be divided into four subsystems which are Upper I, Upper II, Lower III, and Lower IV, by the aquiclude of mudstone and sandy mudstone. Given the thick aquiclude weakened the groundwater vertical relationship and inter-aquifer flow, it would be beneficial for the suppression of vertical leakage; however, there is a disadvantage when using effective storage space [44].

### 4.4. Regional Volumetric and Dynamic Storage Capacity of the Study Area

The previous performance of the MC-1 well indicated that 1.347 Mm^3^ of mine water had been injected and stored in LFm, with the wellhead pressure always fluctuating around 8.0 MPa. The empirically evaluated radius of influence was 635 m, and the specific mine water storage capacity of rocks near the MC-1 well was 1.06 Mm^3^ per 1 km^2^. According to the volumetric calculation method [45], the following equation was applied:(3)$$V=\eta \cdot \pi \cdot R^{2}\cdot M\cdot S$$
where V is the dynamic storage capacity of LFm (m^3^), R is the influenced radius by mine water injection (m), M is the thickness of LFm (m), S is the specific storage (m^−1^), η is the empirical value (dimensionless) for storage capacity, ranging from 0 to 1.

The total volumetric of LFm was evaluated at 3,103.77 billion m^3^ in the study area. Based on the proportion of confirmed sandstone aquifers in LFm of the MC-1 well was 36.1% and the specific porosity of pore diameter over 10 μm was 0.0224, the estimated dynamic storage capacity was 25.10 billion m^3^ for the total area of 106,240 km^2^ in this study area. The calculated volume of 2.36 Mm^3^ per 1 km^2^ in the study area was found to be larger than the measured specific data of the MC-1 well. This indicates that the potential storage capacity of LFm may be higher than expected, so more pre-treated mine water could be transferred and disposed in the deeper aquifer through the projects in coal mines. As shown in Figure 3b and Figure 4c, the local zone with structure development in the dominant W-E and NE-SW directions would be more suitable.

### 4.5. Deep Groundwater Flow Patterns Revealed in Cross-Sections Induced by Mine Water Injection

The deep groundwater regime within the large hydrogeological basin is controlled both by geology and topography [46]. Studying its flow patterns proved to be the ideal method for understanding the groundwater flow process, paths, direction, system, circulation and geochemical evolution [47]. The large-scale sedimentary Ordos Basin was developing with the nested groundwater flow systems, such as the regional, intermediate, and local-scale groundwater systems or subsystems [48]. Regional groundwater flow patterns, hydrodynamic trends, geologic setting and evolution of the study area controlled the feasibility of groundwater reinjection [49]. It was critical for the development of mine water management, especially in the groundwater depression cone formed as a result of overexploitation in coal mining, and the recharge of mine water injection that affected the precipitation-dissolution balance in minerals [50]. Generally, the undulated topography-driven inflow theory proposed by Tóth could be responsible for characterizing the groundwater baseflow pattern in the study area [51] due to the limited geologic and hydrogeologic knowledge [52].

Similarly, 2D simulations of W-E and S-N cross-sectional models through the basin-scale in the study area, as shown in Figure 4a, were selected as the target to investigate deep groundwater flow. The SEEP2D module in GMS 10.1.4 (Groundwater Modeling System) was introduced to simulate and classify different catchments. The specially stratified structure of the sedimentary Ordos Basin controlled the deep groundwater flow pattern. With the land surface as the recharge source, its topographic elevations were equal to the water table, therefore, the undulating elevation of water-table could be assigned by the nodes’ elevation of land surface. So, the surface was set as special head boundary and the each node elevation was equal to the special head [53]. The bottom boundary and two vertical boundaries of cross-sectional model were configured as no flow boundary condition. The conductivity of LFm was much poorer than its overlying formations. The groundwater regime was obtained from Equations (2) and (3).

Regional basin-scale hydrology was generally characterized by gravity-driven flow and considered as steady state. The hierarchically and steadily developed nested flow systems of basin asymmetry [54] for W-E and S-N cross sections in the study area are presented in Figure 6. The groundwater flows from the west to the east and from the north to the south, which is correlated with the overall terrain gradient [55] and verified by the Tóth basin theory. The groundwater-fed and/or salt lakes, flowing wells and topographic lows all became the indicators of discharge zones. The configurations of head isolines within four catchments in the W-E cross section enables the contouring of local subsystems and the delineation of regional groundwater flow patterns. The Tuwei River and Kuye River were the discharge zones in the study area. The groundwater from the western highland eventually flowed into the eastern catchment of the Kuye River [56]. A previous study evaluated that 71% of runoff reduction was induced by coal mining in the Kuye River basin [57], which was the discharge zone of subsystem and was easy to cause mine drainage during mining because water-rich aquifers were not conducive to mine safety production.

The regional hydrodynamic regime of LFm was in the focus of this study aimed to evaluate the potential capacity and influence resulting from mine water injection activity. The overlaying low-permeability sandstones and mudstones aquitards limit the groundwater flow between the shallow and LFm systems. Deep groundwater in LFm passes and flows slowly, sometimes is under a nearly stagnant state, toward its outcrop in the Kuye River and Tuwei River, which were also the discharge zone of regional LFm groundwater. Therefore, the higher heads in the western recharge area than the eastern lower discharge area constituted the driving force for the groundwater circulation in LFm aquifers, but the nonlinear relationship between topography and the water table might have been influenced by local variability in epeirogenetic geology and climate [58,59].

The S-N cross-section with the larger extension than the W-E cross-section has the groundwater flow regime that can be described by the same classical gravity-driven model with local, intermediate and regional patterns [60], as shown in Figure 7. The Wuding River, Dali River, Mahe River and Yanhe River served as the discharge areas of the regional natural hydrogeologic model [61]. The highland in the northern area was recharged the southern Wuding River catchment. Groundwater from the high water-table zone of the Mahe River is discharged to the Yanhe River and Wuding River, respectively.

We found that coal mining activities, including massive drainage, pumping and mine water injections, generated new flow subsystems which covered capturing zones in the coal measures of mining area (depression cone) and releasing zone in the deep mine water injection aquifers (groundwater mound), as shown in Figure 7. The new flow subsystem influenced the regional groundwater flow direction and slightly changed the basin-scale flow pattern [62]. When the coal mining activities became basin-scale and multiple, more internal catchments and flow subsystems could potentially appear. This would be a fundamental factor in describing flow patterns under the influence of mine water injection.

As for mine water injection activity around coal mining claims (the MC-1 well), the emergence of streamlines clearly showed that the groundwater in shallow Jurassic aquifers was pumped and drained to the surface. Extensive coal mining resulted in the appearance of the new fractures generation in overlying rocks. With the influence of complicated water-rock (coal) interaction, water mixing, microbial, geothermal and human activities, groundwater quality was obviously affected when water flowed along those fractures into the panels, roadways and eventually became the mine water. Large basin-scale mine water drainage prompted the downtrend of the regional water table, especially many local depression cones and subsystems of groundwater were distributed over coal mines.

Mine water injected into the tight and low permeability LFm aquifer may move downdip into the deeper zone and the Wuding River [63] in the long-term, with a groundwater mound with high pressure that exists the around coal mine. Generally, the streamline could flow to upward formations if faults of high permeability occurred [64]. Vertical mine water migration through a sequence of low porosity and low permeability layers into shallow aquifers was extremely unlikely in the stable cratonic basin. The groundwater mound obviously changed the groundwater circulation among upstream, downstream, and stored water [65].

### 4.6. Long Term Evolution of the Groundwater Flow Field

The LFm confined aquifer boundary was set as the weak-flow condition due to low permeability and the characteristic of regional-scale low porosity. Laterally extensive low-permeable layers were particularly common in the Ordos basin [66]. Most of groundwater in the study area of LFm circulates in a stagnated movement [67]. In contrast, mine water injection created the artificial recharge source. To understand the scale of these perturbations, we developed the hydrogeological conceptual model with the layered system at GMS by the topography-driven regional groundwater flow method. The parameter configuration included hydraulic conductivity, specific storage, and porosity with the values of 0.011 m/d, 0.000013 and 0.05 for the targeted simulation zone of the Hujierte and Nalinhe mining regions, where were assumed that the hydraulic fracturing had changed and enhanced the hydraulic conductivity to 0.011 m/d. This *K* value was obtained from the repeatedly pressure decaying tests of MC-1 well. Porosity changes induced by the pressure increase were not considered. The specific storage was inversed by the observation well of MC-1. The other region parameters were valued at 0.0001 m/d, 0.000009 and 0.05; they were obtained from the lab test, original groundwater table recovery test and high pressure of hydraulic fracturing.

The natural and original region-scale groundwater field of LFm was presented in Figure 8a. The discharge zones with a lower elevation consisted of the Yellow River catchment [68] with its distributaries: the Wulanmulun River, Kuye River, Tuwei River, Wuding River, Yanhe River and Luohe River. Our objective was to scrutinize the evolution process of mine water injection within 1 year at the rate of 25,000 m^3^/d for each well. The long-term evolution of different groundwater fields is shown in Figure 8b. The hydraulic fracturing process and vertical leakage through the caprock were not considered in this simulation due to the restriction and constraint from the stable, thick and stratiform rocks in Ordos Basin. The process of hydraulic fracturing was important to clarify the environmental influence and groundwater evolution; it was a complex and challenge topic. The vertical leakage of mine water through borehole well, large faults and fault re-activation from the injection activities would cause more serious groundwater pollution than the directly surface drainage. This paper concentrated on the ideal groundwater flow in the stratiform LFm, which was fractured by the continuous mine water injection with the bottom hole pressure up to 32.0 MPa.

Low groundwater velocity and the rock properties of LFm constrained the migration scope of mine water injection, which was similar to the CO_2_ Capture, Utilization and Storage Program in Shenhua experimental site in Ordos Basin. Relatively low porosity and permeability LFm was favorable for fluid storage due to the wellbore storage effects would clearly separate the hydraulic fracturing region and originally undisturbed LFm. Therefore, mine water injection activities in Hujierte and Nalinhe coal mining regions had significantly increased the water pressure, flow velocity and altered local groundwater subsystem for LFm. The external region around the above coal mining region was not showed high groundwater contours. Only the groundwater gradient zone with high dense isolines was distributed between them. With the continuous constant injection rate at the initial period of 1 month, the system response of groundwater pressure to injection is very rapid, with the notable increases in groundwater contour [69]. The characteristics of local groundwater head contour were similar to the regional groundwater gradient because mine water was filling with the pores and fractures [70]. The fractures resulting from the continuous hydraulic fracturing by mine water injection and original fissures generated by the tectonic stress and movement together became the main flow conduits of the mine water. The velocity of mine water flow along these fractures was obviously higher than the migration through pores. The dual fracture porous medium model [71] applied in LFm can be employed for simulation of mine water injection activities. After a period of 6 months, the relatively thin LFm in the north of Hujierte caused the wellhead water table to be over 2750 m close to these injection wells. Its groundwater contour morphology was found to be in negative correlation with the thickness of LFm. As the time was longer than 1 year, all the wellhead of each grid reached its maximum. The wellhead water table ranged from 2050 m to 4200 m in Hujierte and Nalinhe coal mining regions. When the mine water injection terminated, the water pressure observation of each well injection was resumed, and the observation table lines are shown in Figure 9.

The groundwater level of the deep part of LFm reached the steady condition before injection activity, as shown in Figure 9. When mine water injection started, the recovery of groundwater head for wells were initially up to the maximum within 1 year. Thereafter, a slight decrease in groundwater head followed for the instants of 2 years, 9 years, 29 years, 49 years, 99 years, 299 years and 499 years after injection, which was different from the random injection rate fluctuations on pressure and geomechanical stress [72]. Using the logarithmic coordinate of temporal dimension allowed us to find out that the groundwater head could not recover to its initial baseline within 500 years. The property of low porosity and permeability constrained the extent of the groundwater head in Hujierte and Nalinhe coal mining regions. As the scenario in Figure 8b shows, the dense isoline area of the groundwater head was distributed around the boundary of coal mining claims and was difficult to extend outward. Therefore, it was beneficial to the geological storage of mine water in the basin-scale for the Ordos basin, but the renaissance of induced seismicity should be avoided [73]. The high frequency of water injection often induced the seismic activities around wells and was analogous to faults re-activation seismic in the gas extraction [74].

Simulation results suggested that the proper injection management and concept would significantly reduce the unwanted pressure variation and associated effects [75]. The monitoring of pore-fluid pressure for induced seismicity was critical for the mine water injection activity in the long-term period.

## 5. Conclusions

Mine water is not only the unconventional groundwater resource, but also can be reserved in the deep saline aquifers for emergency. The arid and semi-arid region of the Ordos basin in China was menaced by the massive drainage of mine water and scarce groundwater resource. In contrast to the current mine water treatment techniques and other methods, the new and novel approach of mine water injection and storage has significant advantages, and is able to alleviate the currently grim situation as the scientific in-situ experiment of mine water injection in the area of LFm succeeded. The mine water injection activity by coal enterprises would be effectively operated in the study area, but has been poorly studied. Hence, studying the regional deep groundwater pattern was quite necessary for the management and monitoring of mine water injection and groundwater quality evolution. Geophysical technique of gravity and magnetic anomaly, 37 deep wells, logging data of MC-1 well, cross-sectional and numerical models were all combined and focused on revealing the hydrogeological characteristic and deep groundwater flow pattern induced by mine water injection activity.

The main conclusions and findings of this study can be summarized as follows:

The top and bottom interface of LFm were similar to the terrain features due to the stable Yishan Slope in the sedimentary Ordos Basin. The characteristics of thickness and local groundwater systems were mainly influenced by the dominant W-E and NE-SW directions of faults as it follows from studying the geophysical gravity and magnetic anomaly. The horizontal heterogeneity was helpful for justifying well site selection.

Geological logging of MC-1 well revealed more than 35 sub-aquifers in LFm, with 38.1% of its thickness being considered as aquifers. The vertical distribution for the aquiclude of mudstone weakened the vertical leakage and inter-aquifer flow in LFm. The majority of relatively dry sandstone was able to be transformed into effective aquifers during mine water injection and enhanced the storage space for water.

As a result of studying regional volumetric and dynamic storage capacity the calculated volume of 2.36 Mm^3^ per 1 km^2^ was set as one of the criteria for well design. The estimated mine water storage capacity was 25.10 billion m^3^ in this study area. The analysis of two hydrogeologic cross-section models in basin-scale allowed us to suggest that the regional groundwater flow is directed from the west to east and from the north to the west; the lower elevation zone of the rivers’ catchment is located in the discharged region. Compared with the previous literature, the hierarchically and steadily nested flow systems affected by coal mining activity provide insights into the dynamics of basin-scale in the study area.

Mine water drainage in the shallow aquifer formed the groundwater depression cone whereas mine water injection in LFm generated the groundwater mound located in the coal mining claims with a low ability to migrate further. The groundwater simulation based on the flow field calculation revealed that the recovery of groundwater head slightly decreased and the head is not expected to recover to its initial baseline within 500 years. This was beneficial to the geological storage of mine water and management for coal enterprises, as well as the fundamental findings to storage mine water and CO_2_ collaboratively.

## Figures and Tables

**Figure 1 ijerph-19-15438-f001:**
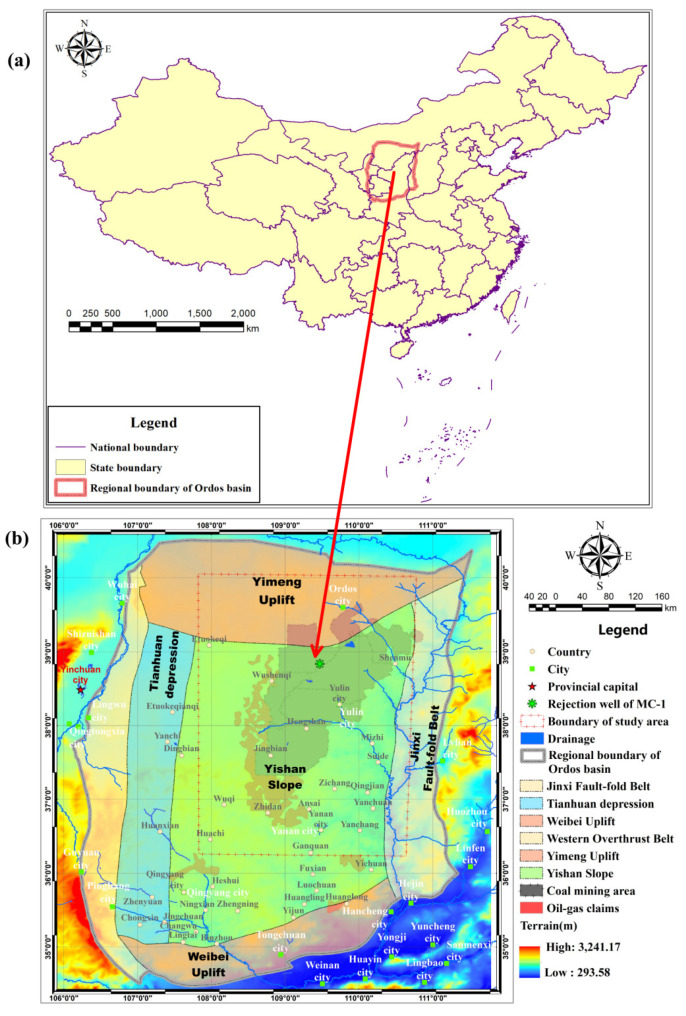
(**a**) The location for the Ordos Basin and the study area, (**b**) coal mining claims distribution and geological structure in the study area.

**Figure 2 ijerph-19-15438-f002:**
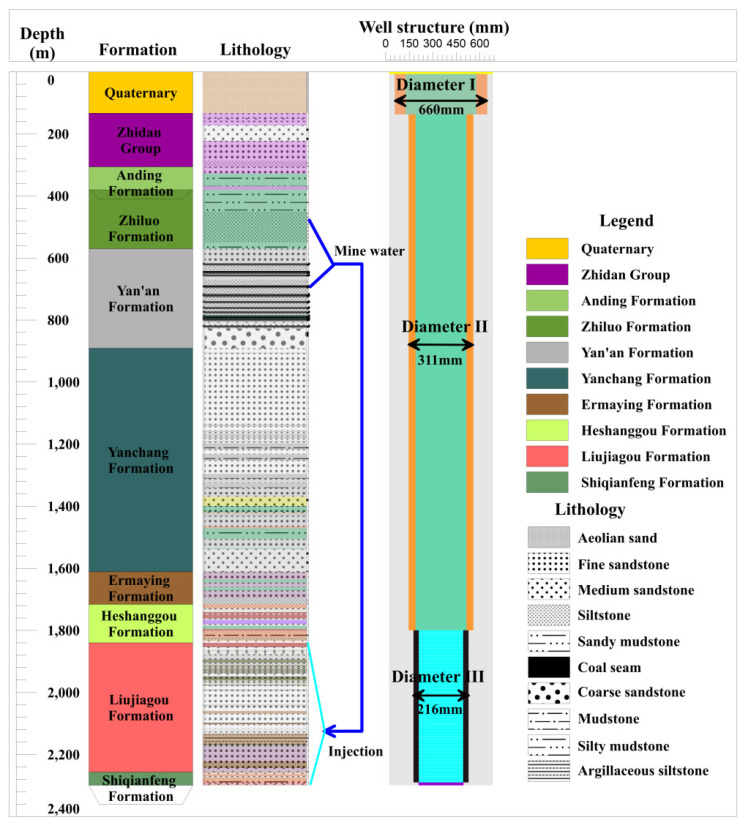
Ordos Basin hydrostratigraphy with mine water injection well structure. Mine water from the coal measure strata named Yan’an Formation is injected into Liujiagou Formation. Different lithology with colors is showed.

**Figure 3 ijerph-19-15438-f003:**
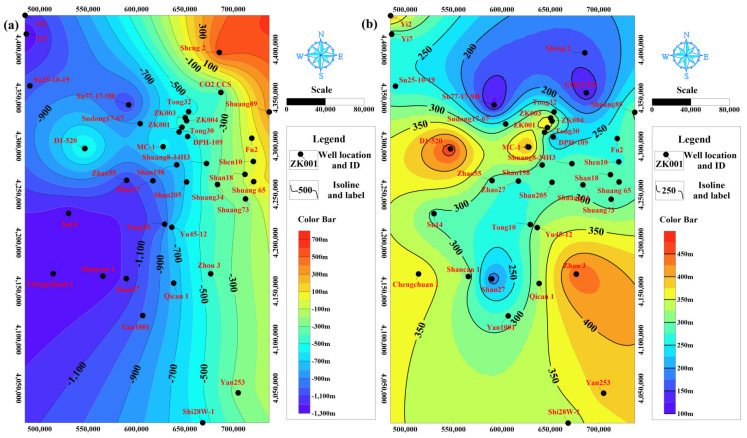
(**a**) The top boundary elevation of LFm. (**b**) The thickness isoline of LFm in the study area. The identified 37 wells were located at the eastern margin of the Ordos basin.

**Figure 4 ijerph-19-15438-f004:**
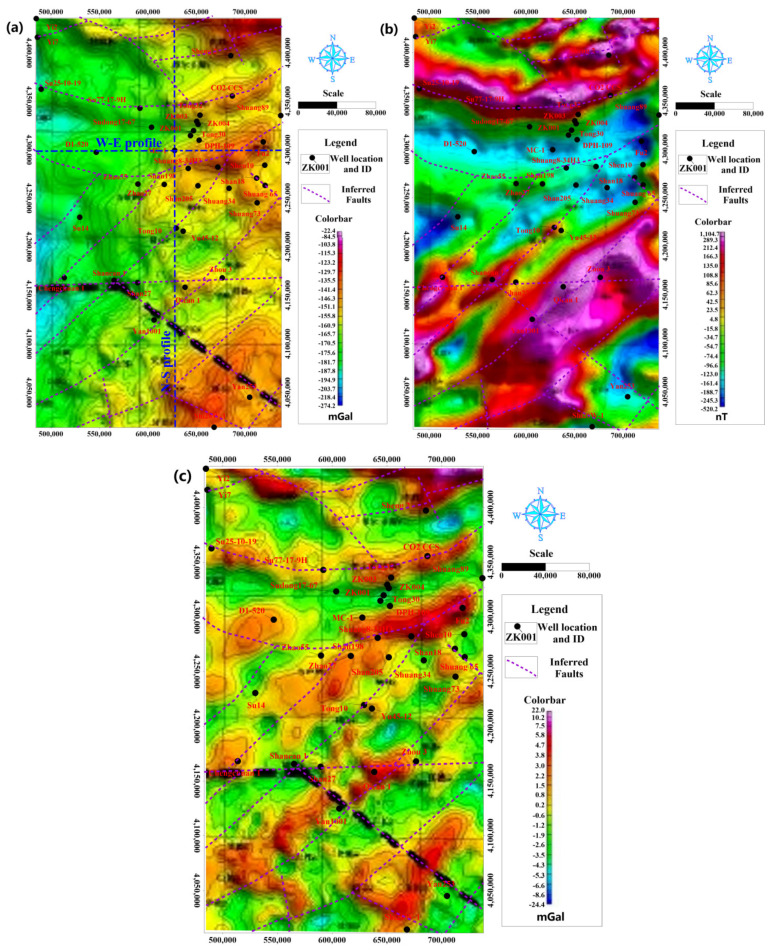
(**a**) Bouguer gravity anomaly, (**b**) Reduction-to-the-pole of magnetic anomaly, (**c**) Small-scale of residual gravity anomaly for the study area.

**Figure 5 ijerph-19-15438-f005:**
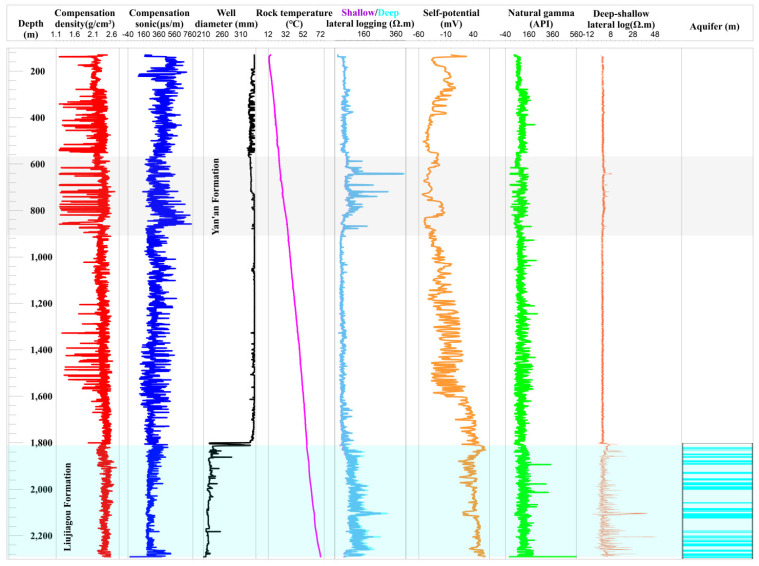
Geophysical logging with the various methods of compensation density, sonic, diameter, temperature, shallow/deep lateral logging, self-potential, natural gamma and deep-shallow difference of lateral logging for the MC-1 well in Muduchaideng coal mine. The aquifers distribution in LFm is shown in the right column.

**Figure 6 ijerph-19-15438-f006:**
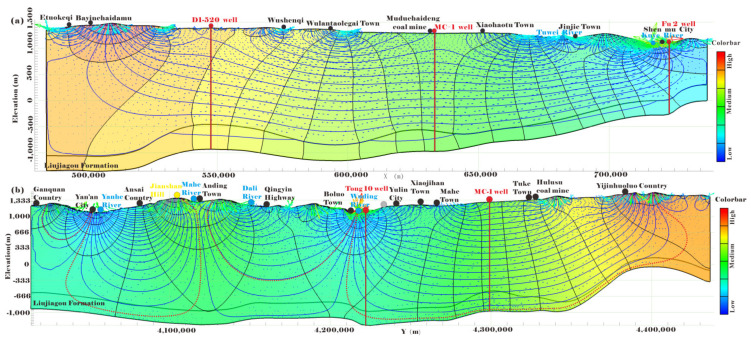
Regional groundwater head and flowline for (**a**) W-E and (**b**) S-N cross-section. The blue line was the groundwater flowline and indicated the groundwater moving path. The small arrow with rainbow color was described the direction and velocity magnitude of regional groundwater. The red arrows were located near the shallow surface or rivers, it is the zone of intensive groundwater circulation. The blue arrows were almost located in the deeper aquifers with the very low flow rate. The black contour shows the groundwater head.

**Figure 7 ijerph-19-15438-f007:**
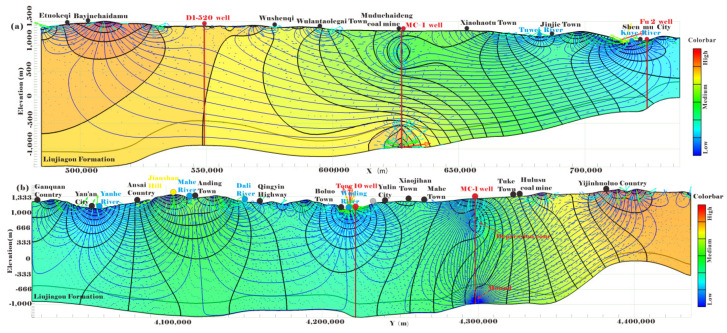
Regional groundwater head and flowline change induced by mine water injection activity through the MC-1 well for (**a**) W-E and (**b**) S-N cross section. Mine water drainage in shallow aquifers and water injection activities in deeper strata around the MC-1 well were simulated to manifest their effect on the regional circulation of groundwater.

**Figure 8 ijerph-19-15438-f008:**
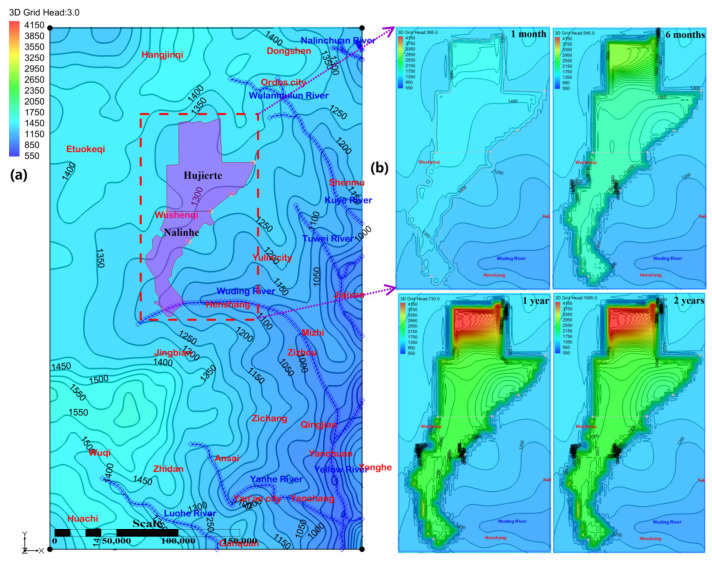
(**a**) The simulated original hydrodynamic field of LFm by the topography-driven regional groundwater flow method. Hujierte and Nalinhe coal mining regions were selected as the large-scale application sites for mine water injection. Each injection well interval was set 3 km × 3 km and the dotted rectangle was the most sensitive zone of water heading. (**b**) Water-table fluctuation zone of water injection activity of the time step at 1 month, 6 months, 1 year, 2 years, (**c**) 9 years, 29 years, 49 years, 99 years, 299 years and 499 years. It was the magnified zone of the dotted rectangle.

**Figure 9 ijerph-19-15438-f009:**
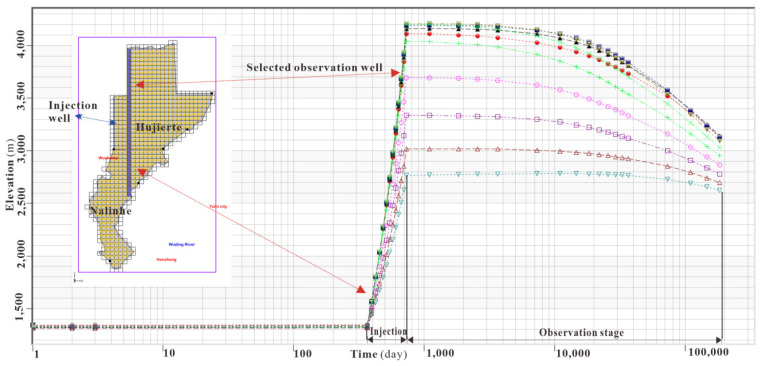
Wellhead water table of each stage in mine water injection activity. The ten observation wellhead table lines are evenly distributed in the selected observation grid row. Different line colors and logarithmic coordinates of x dimension are used to enhance the contrast.

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
