# Peer review of "Deep Groundwater Flow Patterns Induced by Mine Water Injection Activity"

_ijerph, 2022, doi:10.3390/ijerph192315438_

Round 1

Reviewer 1 Report

The manuscript offers an interesting topic, with interesting conclusions. Before considering publication, it is suggested to improve the manuscript to a publishable form. Please make sure that your manuscript fully proves that this work is fundamentally novel. Specific comparisons should be made with previously published materials that have similar purposes. Explain how significant progress has been made in this work. Please ensure that your summary and conclusion not only summarize the main findings of your work, but also explain how this work fundamentally advances the field compared with previous literature.

 Some more specific critiques:

1.      Lines 21, 192, 225, 228, 232, 235, 238, 239, 359, 435 and 436: 2 and 3 are the superscript power number. The m3 should be m3 and km2 should be km2, respectively.

2.      Line 34: Please add a description of the groundwater water quality, mineral type and content in the introduction. The manuscript should also explain the relevant research introduction of hydrogeochemical composition of groundwater in the study area, so as to understand the geochemical evolution of groundwater induced by mine water injection.

3.      Lines 35-41: Please delete these text descriptions related to manuscript editing.

4.      Lines 176-177: Please mark (a), (b), or (c) on the Figure.

5.      Lines 269-270 and 309-310: Please mark (a) or (b) on the Figure.

6.      Line 244: …..would be more appropriate? suitable.  Please further correct the description.

7.      Lines 266-267: The LFm top and bottom boundaries as well as two vertical boundaries were configured as no flow condition. These boundary conditions are cited in reference [48]. However, reference [48] is a hypothetical case. An analytical solution is developed based on the boundary conditions of the hypothetical case in reference [48]. These boundary condition states cannot be directly referenced as boundary conditions set by the SEEP2D module in GMS 10.1.4 (Groundwater Modeling System). The boundary conditions must be reasonably set by investigating the surface and underground environmental conditions and hydrogeological layered structures. The boundary conditions proposed in the current manuscript are unreasonable, and the simulation results could do not conform to the real environment of the site.

8.      Lines 360-361: Hydraulic fracturing and vertical leakage through the cap rock were not considered in this simulation. Why are the two factors not considered? These two factors may be two important environmental conditions that affect the results of the research topic of this manuscript. Please add research analysis or discussion on the simulation scenario of hydraulic fracturing and vertical leakage through the caprock to clarify whether the results of this manuscript can be applied in practice.

9.      Lines 103, 146, 176, 177, 269, 270, 309, 310 and 363: The resolution of the Figure must be further enhanced. The text of the legend’s description must be clearly indicated and displayed.

10.  Line 363: Figure 8 shows the results of the simulated groundwater level, where Figure 8 (a) shows the initial state of groundwater level, and Figure 8 (b) shows the groundwater level at 1 month, 6 months, 1 year, 2 years, 9 years, 29 years, 49 years, 99 years, 299 years and 499 years. Figure 8 (a) The groundwater level in the simulation area is continuous. However, because of the boundary conditions set in the simulation area, there is a completely independent state inside and outside the simulation boundary. This shows that the geological structure of the site in the boundary state is completely impermeable to groundwater. Please further explain its rationality.

11.  Please explain the research methods and results in the manuscript, which fully proves that this work is basically a novel. In addition to describing the case results of other review studies, please put forward the description that the research results can surpass or innovate other past research and development.

Author Response

Dear Reviewer:

Thanks very much for your detailed comments and suggestions of improving the quality of this manuscript. These comments are all valuable and very helpful to the authors. We have read it carefully and improved this manuscript according to the following comment.

Reviewer 2 Report

Overall the manuscript is fine, some highlighted comments are given in the attached reviewed manuscript file to be improved.

Author Response

The authors are very excited to write to express our thanks to your comments and advice for the improvement of this manuscript entitled ”Deep groundwater flow patterns induced by mine water injection activity”. These comments are all valuable and very helpful for revising this paper, as well as very important guiding significance to our researcher team.

Reviewer 3 Report

Many figures contain lettering that is too small for the reader to see.  For example, in figure 8b, the legends that describe the color scheme cannot be read and so interpretation of the figures for the time periods is lost to the reader.

It would seem that another model run at lower injection rates might answer the pressure questions?

Author Response

Thanks for your comments and suggestions for improving the quality of this manuscript. This manuscript showed a research on deep groundwater flow patterns induced by mine water injection activity. The authors had read carefully and improved this manuscript in according to the following comment.

Round 2

Reviewer 1 Report

Dear authors, thank you for addressing my comments.